# Spatiotemporal Distribution of Heatwave Hazards in the Chinese Mainland for the Period 1990–2019

**DOI:** 10.3390/ijerph20021532

**Published:** 2023-01-14

**Authors:** Wei Wu, Qingsheng Liu, He Li, Chong Huang

**Affiliations:** 1State Key Laboratory of Resources and Environmental Information System, Institute of Geographic Sciences and Natural Resources Research, Chinese Academy of Sciences, Beijing 100101, China; 2University of Chinese Academy of Sciences, Beijing 100049, China; 3Jiangsu Center for Collaborative Innovation in Geographical Information Resource Development and Application, Nanjing 210023, China

**Keywords:** heatwave, hazard assessment, heatwave index, spatiotemporal distribution, Chinese mainland

## Abstract

Heatwaves occur frequently in summer, severely harming the natural environment and human society. While a few long-term spatiotemporal heatwave studies have been conducted in China at the grid scale, their shortcomings involve their discrete distribution and poor spatiotemporal continuity. We used daily data from 691 meteorological stations to obtain torridity index (TI) and heatwave index (HWI) datasets (0.01°) in order to evaluate the spatiotemporal distribution of heatwaves in the Chinese mainland for the period of 1990–2019. The results were as follows: (1) The TI values rose but with fluctuations, with the largest increase occurring in North China in July. The areas with hazard levels of medium and above accounted for 22.16% of the total, mainly in the eastern and southern provinces of China, South Tibet, East and South Xinjiang, and Chongqing. (2) The study areas were divided into four categories according to the spatiotemporal distribution of hazards. The “high hazard and rapidly increasing” and “low hazard and continually increasing” areas accounted for 8.71% and 41.33% of the total, respectively. (3) The “ten furnaces” at the top of the provincial capitals were Zhengzhou, Nanchang, Wuhan, Changsha, Shijiazhuang, Nanjing, Hangzhou, Haikou, Chongqing, and Hefei. While the urbanization level and population aging in the developed areas were further increased, the continuously increasing heatwave hazard should be fully considered.

## 1. Introduction

Global warming has contributed to the increasing frequency and intensity of extreme climate disasters [1,2], and heatwaves account for the highest mortality rate among all extreme disasters [3,4]. The 2003 European heatwave caused approximately 70,000 deaths [2,5,6], and at least 5758 heat-related deaths were caused by the 2013 Chinese heatwave [7,8,9]. The 2019 Indo-Pakistani heatwave caused at least 400 deaths, accompanied by widespread drought and water scarcity [10], and it was one of the factors that led to the Bihar encephalitis outbreak [11]. Furthermore, the 2021 heatwave in Western North America led to a record temperature of 49.6 °C in Canada [12], resulting in more than 1400 deaths [13,14]. Previous studies have shown outdoor workers, the poor, and the vulnerable to be more vulnerable to the effects of heatwaves, as well as the agricultural and forestry disasters caused by heatwaves [15,16,17]. Therefore, research on heatwaves is imperative in the context of global warming.

A heatwave is a weather process involving high temperatures, high humidity levels, and long durations, which leads to bodily discomfort and may threaten public health and safety, increase energy consumption, and affect social production activities [18]. At present, various temperature indices are available to define heatwaves, such as the air temperature [19,20,21], surface temperature [22], and dew point temperature [23]. There is a growing tendency for analyzing heatwaves using apparent temperature indices such as the physiological equivalent temperature (PET) [24], standard effective temperature (SET) [25], temperature–humidity–wind index (THW) [26], humidex [27], wet-bulb globe temperature (WBGT) [28], universal thermal climate index (UTCI) [29,30], and heat index (HI) [31], which take into consideration additional indicators such as the relative humidity, wind speed, and atmospheric precipitation. Notably, no consistent definition exists regarding the temperature threshold and number of days of a heatwave [32]. The results of a previous study showed that the HI is a valid index for predicting heatwave weather [31], which has been used by many researchers in the study of heatwaves in China [33,34,35]. Therefore, the HI was used to identify and define the heatwave index (HWI) and heatwaves. Previous studies have shown that the adaptation of populations to climates varies from region to region [36,37,38,39], which means it is rather unreasonable to judge a heatwave in a complex climate region based on a single absolute threshold or a simple relative temperature threshold.

In the Special Report on Managing the Risks of Extreme Events and Disasters to Advance Climate Change Adaptation [40] and the Fifth Assessment Report (AR5) [41] of the Intergovernmental Panel on Climate Change (IPCC), a risk-centered assessment framework was presented, in which risk was expressed as a function of hazard, exposure, and vulnerability [42]. Therefore, clarifying the spatiotemporal distribution of heatwave hazards is a necessary prerequisite for judging the evolutionary trend of heatwave risks. In the given framework, hazard was defined as the external factors of a system that pose a serious threat to the system [43]. Some researchers have directly used the temperature index to define hazards [29,42]. However, the scientific literature shows that a heatwave hazard is a measure of the severity of heatwave events, usually determined by the intensity, duration, frequency, and extent of heatwaves [32,44,45,46,47,48,49] and calculated using the graphic overlay method [50,51,52,53].

China is a sensitive and significant area for the impacts of global climate change [54]. Since the 21st century, heatwaves have frequently occurred in China, and heatwaves are expected to form the new normal of the country’s summer weather in 2030 [55]. In addition, the results of Sun’s study on Chinese heatwaves showed a linear increase in intensity and a significant increase in frequency [56]. Early studies involving heatwave hazard assessments in China used different definitions based on the meteorological stations considered and focused on small areas and short time periods, resulting in the shortcomings of a discrete distribution and poor spatiotemporal continuity. For instance, Zhao et al. used the weighted averages of the surface temperature, air temperature, and air pressure to assess the hazard level in Ningxia for the years 2014–2019 [57]. Wang defined heatwave hazards based on the interpolated data from meteorological stations. The duration, intensity, accumulated heat days, and length and intensity of the return period were used to analyze the heatwave hazard level of the Yangtze River Delta region [58]. Zhan et al. used 35 °C as the temperature for a heatwave and analyzed the heatwave characteristics, such as the intensity and duration, of the North China Plain on a 0.5° grid scale interpolated by the maximum temperature data. Relatively few long-term studies of Chinese heatwave hazards have been conducted with a finer grid unit scale and without the use of relative heat thresholds [59].

Since the 21st century, the intensity and extent of Chinese heatwaves have continued to increase, causing serious impacts on residents. For the present study, heatwaves were defined based on a HWI calculated using the torridity index (TI, °C) and HI, and an annual heatwave hazard dataset with a spatial resolution of 0.01° was built based on the daily monitoring data from 691 meteorological stations from May to September 1990–2019 in the Chinese mainland (hereinafter referred to as China). The spatiotemporal distribution of heatwave hazards in China was further evaluated for this period, and the average hazard (AH) trends at the city and county levels were determined.

## 2. Materials and Methods

### 2.1. Study Area

Spanning almost 50 degrees in latitude, China has a wide variety of climate types. The summer temperatures are consistently high across the country, and the temperatures gradually decrease as the latitude increases. Generally speaking, summer in China spans the months of June–August. However, in some cities and regions, the temperature remains high in May and September, affecting 18% of the total population, since China’s population is very large compared to other countries [55,60,61].

### 2.2. Data Sources

Maximum temperature (MT, °C) and average relative humidity (RH, %) data from 699 meteorological stations in China (Figure 1) for the years 1990–2019, which were obtained from the surface climate daily value dataset (V3.0) of the China Meteorological Data Network (http://data.cma.cn/, accessed on 23 January 2021), were quality-controlled and used to calculate the TI and HI. We focused on the extended summer period covering the months of May–September to account for the impacts of early and late summer heatwaves [62]. Four sets of random validation stations were selected to make their distribution more uniform. Containing 10% of all stations (of a total of 69), each set randomly selects one day of the month to apply, for a total of four days per month using only interpolation points for interpolation. Some random validation stations overlap, so we obtained a validation set of 241 points and a training set of 450 points.

Data on the elevation and geographical and administrative divisions were obtained from the Resource and Environmental Science and Data Center of the Chinese Academy of Sciences (https://www.resdc.cn/, accessed on 10 January 2021). The geographical divisions used in this study were as follows: north (N) consisted of Beijing, Tianjin, Hebei, Shanxi, and Inner Mongolia; northwest (NW) consisted of Xinjiang, Qinghai, Gansu, Ningxia, and Shaanxi; northeast (NE) consisted of Heilongjiang, Jilin, and Liaoning; southwest (SW) consisted of Tibet, Yunnan, Sichuan, Chongqing, and Guizhou; south central (SC) consisted of Henan, Hubei, Hunan, Guangdong, and Guangxi; and east (E) consisted of Shandong, Anhui, Jiangsu, Shanghai, Zhejiang, Jiangxi, and Fujian (Figure 1). The data on administrative divisions involved 34 provincial administrative units, 371 municipal units, and 2902 county units.

Based on the commercial resource concentration, city hubs, city residents’ activities, lifestyle diversity, and future plasticity, the RISING Lab constructed an index model of cities’ business attractiveness and divided 337 cities into six categories: four first-tier cities, 15 new first-tier cities, 30 second-tier cities, 70 third-tier cities, 90 fourth-tier cities, and 128 fifth-tier cities [63]. To investigate the heatwave hazards in cities at different levels of economic development, the city ranking data were used in this study.

### 2.3. Data Preparation

#### 2.3.1. Spatial Interpolation of Meteorological Data

At present, the methods for obtaining meteorological data for a region based on the spatial interpolation of discrete meteorological stations mainly include the inverse distance weighting (IDW) [64], Kriging [49], Parameter-Elevation Regressions on Independent Slopes Model (PRISM), trend surface analysis (TSA), and thin plate smoothing spline (TPS) [65] methods. Among them, Kriging and IDW are the most widely used in practical applications. However, their interpolation accuracies are not high enough for unevenly distributed meteorological data and complex terrains [66]. The TPS method based on the principle of minimum curvature uses the characteristics of smoothly distributed meteorological elements in space to fit the surface, which is more reflective of the natural spatial distribution of things [67]. The Australian National University Spline (ANUSPLIN) is a special meteorological data space interpolation program based on TPS that can effectively simulate terrain. It has been proven to be a reliable software program for meteorological data interpolation [68,69], and its interpolation accuracy in complex terrain areas is better than that of other methods [70,71]. It can complete the spatial interpolation of more than two surfaces at the same time, so it is especially suitable for the interpolation of time series meteorological data [72,73].

Thus, in this study, we used the ANUSPLIN to interpolate MT and RH with the help of a one-dimensional independent covariate (the elevation data).

#### 2.3.2. Interpolation Accuracy Validation

The accurate interpolation of MT and RH forms the premise for calculating heatwave properties. Therefore, the cross-validation method was used to verify the MT and RH datasets, and the process was as follows.

The mean absolute error (MAE), normalized mean absolute error (NMAE), root mean square error (RMSE), and normalized root mean square error (NRMSE) values were calculated for the units of different timescales (day, month, year) using the validation set results as the true values and the interpolated results as the predicted values.

The predicted values were fitted as independent variables and the true values as dependent variables, while the slope, R^2^, and *p* values were calculated for the different timescale units.

A total of 18 cross-validation results for the two variables—i.e., MT and RH—were selected according to the descending order of R^2^ at the different timescales for presentation. The numbers selected were for day 4, month 3, and year 2.

The results in Table 1 show that the minimum MAE values for the MT and RH interpolations were 0.80 °C and 3.45%, respectively. The minimum RMSEs were 1.03 °C and 4.76%, respectively. The minimum R^2^ value was 0.5318, but 98.08% of the results were not lower than 0.7, indicating that the prediction values were highly correlated with the true values (*p* < 0.001). The verification results showed that the interpolation results based on ANUSPLIN were accurate and reliable. Therefore, the MT and RH datasets could be used to calculate the TI and HI.

Table 1 also presents the 18 cross-validation results obtained for the two variables, where 0.98 < slope < 1.0, R^2^ > 0.9, and the results passed the significance test (*p* < 0.01), as well as the NMAE and NRMSE values calculated using the max–min normalization method.

### 2.4. Methods

#### 2.4.1. Definitions of HWI and Heatwaves

The TI mainly considers the influence of the temperature and relative humidity on human comfort [74], which is a heat stress metric similar to the commonly used heat index [75,76,77]. The HI in this study uses the relative and absolute thresholds to define a heat day and considers the cumulative effect of the adjacent days’ TI values [31]. Considering the long durations of high-temperature heatwaves, in this study the HWI of a heatwave was defined as the average value of the daily HI, while a heatwave was defined as an event with TI ≥ TI′ for three consecutive days or more and HWI ≥ 2.8. The HWI threshold was based on a comprehensive HI finding, indicating that the heat grade of the day reaches the heatwave standard if HI ≥ 2.8. The calculation model is given below:(1)$$\{\begin{array}{c} \mathrm{TI}=1.8\;\times \;\mathrm{MT}\;-\;0.55\;\times \;\left(1.8\;\times \;\mathrm{MT}\;-\;26\right)\;\times \;\left(1\;-\;0.6\right)+32,\;\mathrm{RH}\;\leq \;60\%\\ \mathrm{TI}=1.8\;\times \;\mathrm{MT}\;-\;0.55\;\times \;\left(1.8\;\times \;\mathrm{MT}\;-\;26\right)\;\times \;\left(1\;-\;\mathrm{RH}\right)+32,\;\mathrm{RH}\;>\;60\% \end{array}$$
(2)$$\mathrm{HI}=1.2\;\times \;\left({\mathrm{TI}\;-\;\mathrm{TI}}^{\prime }\right)+0.35\overset{N}{\underset{i=1}{\sum }}{1/\mathrm{nd}}_{i}\left({\mathrm{TI}}_{i}{\;-\;\mathrm{TI}}^{\prime }\right)+0.15\overset{N-1}{\underset{i=1}{\sum }}1/{\mathrm{nd}}_{i}+1$$
(3)$$\mathrm{HWI}=\overset{N}{\underset{i=1}{\sum }}{\mathrm{HI}}_{i}$$

TI′ is the threshold of the TI; TI_i_ represents the TI of the i-th day before the current day; nd_i_ represents the number of days from the i-th day before the current day to the current day; HI_i_ represents the i-th day in a heatwave event; N represents the duration of the heatwave event, measured is days (d).

The TI′ was calculated from the quantiles of the TI. First, samples (MT and RH) with MT > 33 °C were selected from the daily meteorological data, and the TI sequence of the samples was arranged in ascending order. The 50th quantile was selected as the local TI′. The calculation method is given below:(4){^Qi(p)=(1 − γ)X(j)+γX(j+1)j=int(p × n+(1+p)/3)γ=p × n+(1+p)/3 − j

Here, ^Qi(p) is the i-th quantile value; X is the TI sequence in ascending order; p is the quantile (0.50); n represents the total number of sequences; j is the j-th sequence number.

First, daily TI raster datasets were determined based on the interpolated MT and RH, and then the TI′ raster was calculated based on Equation (4). Following this, TI′ was used for the comparison data, and each TI was traversed. The obtained value was compared with the value of the corresponding pixel in the comparison data. A pixel value greater than the comparison data indicated heat and was marked as 1, while the opposite was marked as 0, resulting in daily raster datasets of 0 and 1, for a total of 4590. The HI dataset was calculated based on the judgment dataset. According to the definition of a heatwave, the pixels with a single heat event (without interruption) comprising less than three heat days or HWI < 2.8 were ignored to obtain the HWI dataset. Thus, the pixels in the heatwave represented the HWI of the heatwave, the corresponding pixel of a single day with the highest HI in a single heatwave was marked as the HI of that day, and the pixel marked as 0 represented no heatwave occurrence.

#### 2.4.2. Heatwave Hazard Assessment and Classification

A heatwave hazard is usually defined based on the heatwave frequency, intensity, and duration. In this study, the heatwave frequency (HWF), maximum HI of a heatwave (HWMHI), and maximum heatwave duration (HWMD) in a year—representing the frequency, intensity, and duration, respectively—were used to calculate the heatwave hazard. To make the hazard values comparable from year to year, we calculated the maximum value of each indicator for the 30-year period, used this value to normalize the data, and then added the indicators with equal weights, as given in Equation (5):(5)$$\mathrm{Hazard}=(\bar{\mathrm{HWF}\;}+\bar{\;\mathrm{HWMD}}+\bar{\mathrm{HWMHI}})/3$$

Based on the HWI dataset, the data were calculated separately for each year and for each pixel. The heatwaves were judged day-by-day, and the HWF value was increased by 1 in the case of a consecutive heatwave. If there was no value for HWMHI, the maximum value in this heatwave was assigned to HWMHI. If a value already existed, the new HWMHI value was compared with the existing value, and it was used if it was greater than the existing value. If there was no value for HWMD, the number of days for which the heatwave lasted was assigned to HWMD, and if a value already existed, it was compared with the new value and replaced with the new value if the latter was greater than the existing value. This process was followed to obtain the HWF, HWMHI, and HWMD datasets for each year.

The natural breaks method was used to classify the spatial distribution and temporal trends of the AH values, and each was categorized as high, medium-high, middle, medium-low, or low (red section in Figure 2), and as increasing, slightly increased, basically unchanged, slightly decreased, or decreased (green section in Figure 2) [42,50]. Apart from the above two classifications, the hazards in China were further divided into four categories: high hazard and rapidly increasing, high hazard and slightly decreasing or no significant change, low hazard and continually increasing, and others (blue section in Figure 2). The spatiotemporal classification standards of the above four categories are presented in Figure 2. In Figure 2, “or” relationships are present between bands of the same color section, and “and” relationships are present between bands of different color sections; for example, the “high hazard and rapidly increasing” category consisted of the following four scenarios: high hazard and increased, high hazard and slightly increased, medium-high hazard and increased, and medium-high hazard and slightly increased.

The AH values were used to rank heatwave hazards in the Chinese administrative divisions, and the zoning statistics methodology was used in this study. We first obtained the AH means of each administrative division for the 30-year period and then ran a linear regression over time for each administrative region. Following this, we selected first-tier, new first-tier, and second-tier cities and counties and ranked them in descending order for the AH and slope.

#### 2.4.3. Analysis of the Relative Change of Indicators

A time period spanning 30 years was chosen for this study. To better compare the changes in different indicators, the average values for the first five years (FY, 1990–1994) and the last five years (LY, 2015–2019) were considered. In addition, the level of change for each indicator was different for each geographical division. For average TI (ATI), the change also differed each month. Therefore, the ATI was analyzed in terms of the different geographical divisions and months, and each heatwave indicator was analyzed according to the different geographical partitions. Then, the average values for FY and LY were calculated. Moreover, the concept of relative change (RC, %) was introduced to avoid the one-sidedness of the absolute change amount when comparing different months or divisions [78]. The RC was calculated as follows:(6)$${\mathrm{RC}}_{i,g(,m)}=(\bar{i_{\mathrm{LY},g(,m)}}-\bar{i_{\mathrm{FY},g(,m)}})/\bar{i_{g(,m)}}$$

Here, i, g, and m represent the different indicators (ATI, HWF, HWMHI, HWMD), different geographical divisions (N, NW, NE, SW, S, E), and different months (May, June, July, August, September), respectively; $\bar{i_{\mathrm{LY},g(,m)}}$ and $\bar{i_{\mathrm{FY},g(,m)}}$ represent the average values of LY and FY for an indicator, respectively; $\bar{i_{g(,m)}}$ represents the average indicator value for the period 1990–2019. RC_i,g(,m)_ > 0.8 means that the indicator increased sharply for the geographical zone considered in a specific month (if the RC is calculated).

## 3. Results

### 3.1. Dynamic Changes in TI in Different Months

Figure 3 shows the ATI values for the five months considered over the 30-year period. In terms of the TI, the high-value areas (HAs) in China were mostly located in the northwest, east, and south, while the low-value areas (LAs) were located in the SW and NE. With the northward movement of the direct sunlight region and the influence of topography, the HAs expanded in East Xinjiang, Southeast Tibet, and Hainan in May (Figure 3a). After various regions reached high levels with different coverage rates in June, July, and August (Figure 3b–d), the HAs begun to move southward in September (Figure 3e). By this time, apart from the HAs in East Xinjiang, Southeast Tibet, Guangdong, South Guangxi, and Hainan, the TI values for the other areas greatly decreased. The TI in southern China was low because of the rainy season in May and June, so HAs appeared in the north (Figure 3a,b). In July and August, South China began heating up on a large scale.

With the time series as the independent variable and the ATI of each month over 30 years as the dependent variable, linear regression was carried out and the change slope was calculated, with a significance test performed. Figure 4 shows the ATI slopes for the different months and the monthly slope (*p* < 0.05). A slope higher than 0 indicates that the TI had a linearly increasing trend, while a slope lower than 0 indicates that the TI had a linearly decreasing trend. The proportions (slope > 0) were 99.99%, 99.43%, 100%, 99.03%, 98.00%, and 99.92% in the grids passing the significance test, indicating that the trend of increasing TI was widespread and significant. In May, HAs were located in Beijing, Tianjin, Shandong, Yunnan, Sichuan, and other areas (Figure 4a). In June, July, and August, HAs appeared in the central and western provinces of China, East Tibet, West Yunnan, and Inner Mongolia (Figure 4b–d), with a continuous distribution. HAs appeared in the southeastern provinces of China in August and September, as well as in Liaoning, East Inner Mongolia, and Central and West Tibet in September (Figure 4d,e), with relatively discrete distributions. The ATI values dramatically increased in the regions where the slope was greater than 0.1. The monthly slope was reduced by the effects of inter-year changes, but the increasing areas passing the significance test were still consistent with the above results.

Figure 5 shows the annual changes in the national ATI and the RC values of the ATI for different regions and different months. From 1990 to 2019, the ATI was found to increase, but with fluctuations. The minimum ATI value hovered around 110.5 in 1992–1994, while the maximum value appeared in 2007 at about 113.5 °C. The total ATI increase was about 1.99 over 30 years (Figure 5a). The RC of the ATI was the largest (3.39%) in July, and from a regional perspective, the RC values of the ATI in descending order were N > nation > NW > NE > SW > SC > E. The RC values of the ATI in the SW and NE—which are regions sensitive to climate change—were higher than the national average in June and September (Figure 5b).

### 3.2. Assessment of Hazard Indicators

Figure 6 shows the annual averages and slopes for three hazard indicators, namely the HWF, HWMHI, and HWMD, for the years 1990–2019 (*p* < 0.05). HAs for these indicators were mostly found in the NW, E, and SC, followed by the N, while LAs were found in the NE and SW. Extremely high values were found in the southeastern region of the SW (Figure 6a,c,e), while a small number of extremely high values of HWF were found in Southeast Tibet and East Xinjiang. Except for the extremely low values, the HWFs for a large area of the country reached an average of 3–4 times per year and were widely distributed in the eastern provinces of China, Chongqing, and Sichuan (Figure 6a). The maximum HWMHI range in Southeast Tibet was 24–26, the highest values in the country, followed by Chongqing and East Xinjiang at 17–23 and South Hebei, East Henan, Central Zhejiang, South Hunan, Northwest Shandong, and South Shaanxi at 13–20 (Figure 6c). The distribution pattern of the HWMD was the same as that of the HWF. The highest HWMD range (11–16 days) was found in East Xinjiang and Southeast Tibet, with an average of 4–8 days, while the highest HWMD range was about 8–10 days (Figure 6e). In terms of the temporal trend, the percentages of slopes > 0 in areas passing the significance tests for the HWF, HWMHI, and HWMD were 99.84%, 99.98%, and 99.96%, respectively, indicating that the heatwave frequency, intensity, and duration increased in most of the regions where significant changes occurred, such as Southwest Xinjiang, West Inner Mongolia, Chongqing, Henan, Shandong, Jiangsu, Zhejiang, and Hunan (Figure 6b,d,f).

Figure 7 shows the RC levels of the HWF, HWMHI, and HWMD. It can be seen that the RCs of the HWF descended in the order of N > SC > NE > 0.8 > nation > E > NW > SW; the RCs of the HWMHI descended in the order of N > NE > NW > 0.8 > nation > E > SW > SC; and the RCs of the HWMD descended in the order of NW > N > SC > 0.8 > NE > E > SW > nation. All results were greater than 0, meaning the values of all indicators increased in all regions, and each indicator sharply increased in some regions. Here, 2–3 sharply increasing indicators were found in the N, NE, NW, and SC regions. Therefore, a sharper increase occurred in northern China than in southern China. The largest RC was that of the HWMD, which occurred in the NW. The smallest RC was also for the HWMD, which occurred for the nation overall.

### 3.3. Spatiotemporal Distribution of Heatwave Hazard

Figure 8a shows the AH spatial distribution in China for the years 1990–2019. The proportions of low, medium-low, medium, medium-high, and high AH values were 61.07%, 16.77%, 13.07%, 7.06%, and 2.03%, respectively. The areas with AH levels of medium and above were mainly distributed in the eastern and southern provinces of China, Southeast Tibet, East and South Xinjiang, and Chongqing. The low-AH areas were mainly in the Qinghai–Tibet Plateau and the northeastern provinces of China due to their high altitudes and latitudes. Although the rates of increase in these regions were high, they could not reach the heatwave standard in the short term. Figure 8b shows the regression results of the AH values used to assess the annual changes in hazard levels. Except for a few regions, the hazard levels of the regions that passed the significance test all increased to varying degrees, including the eastern provinces of China, South Xinjiang, and Western Inner Mongolia.

The visualization results for the spatiotemporal distribution of heatwave hazards in China are shown in Figure 9. The first category of areas accounted for 8.71% and included Southeast Tibet, South Xinjiang, Chongqing, South Hebei, West Henan, Central Zhejiang, Central and South Jiangxi, and East Hunan. The second category accounted for 0.38%, and the distribution area was concentrated in Northwest Xinjiang, which means that the hazard levels of almost all high hazard areas were increasing. The third category accounted for 41.33%, with a wide distribution area in East, South, and North China, which means that most of the low hazard areas saw an increasing hazard trend that will continue to increase in the future. The spatiotemporal hazard levels of other areas, such as the Qinghai–Tibet Plateau, Yunnan–Guizhou Plateau, and parts of Northeast China, were low due to their high altitudes and latitudes.

### 3.4. Ranking the Heatwave Hazard Levels of Chinese Cities

We evaluated the AH values at the city and county levels and their slopes. The top 100 results at the city and county levels are shown in Table A1 and Table A2, respectively, and the 10 cities and 10 counties with the fastest increases in hazard values are presented in bold. Table A1 shows the AH values of the first-tier, new first-tier, and second-tier cities. Luohe (Henan) was found to have the highest AH, while Suzhou (Jiangsu) had the highest slope (Table A1). Shanghai and the Gulou, Shunhe Huizi, Yuwangtai, Longting, and Xiangfu districts in Henan Kaifeng; the Wancheng District in Henan Nanyang; and the Xiangcheng District in Jiangsu Suzhou were in the top ten cities and counties in terms of fastest increases but not in the top 100 AH rankings, which means the AH values of these regions were not high but their growth rates were at the forefront. The administrative districts that passed the significance test are marked with “*” in the tables (*p* < 0.05), while the proportion of units that significantly increased, as seen in Table A1 and Table A2, equaled 57% and 68%, respectively, at the city and county levels.

Beijing, Chongqing, Tianjin, and Shanghai had relatively low rankings overall (Table 2), but some urban districts in Chongqing had higher rankings at the county level (Table A2). The possible reason for this phenomenon is that the presence of suburbs reduced the statistical significance of the mentioned cities. The top 10 first-tier, new first-tier, and second-tier cities were Jinhua, Zhengzhou, Nanchang, Wuhan, Shaoxing, Changsha, Shijiazhuang, Nanjing, Wuxi, and Changzhou. The top 10 provincial capitals were Zhengzhou, Nanchang, Wuhan, Changsha, Shijiazhuang, Nanjing, Hangzhou, Haikou, Chongqing, and Hefei, and these were termed the “ten furnaces” in this study.

## 4. Discussions

### 4.1. Heatwave Hazards in Chinese Mainland

In 2013, severely high temperatures occurred in South China, resulting in extreme drought and food production reduction [79]. In the summer of 2018, heatwaves in Northeast Asia and East Asia caused more than 43.13 thousand km^2^ of arable land to dry up, and more than 4.6 million people faced shortages of drinking water [80]. Figure 10 shows the heatwave indicators in 2013 and 2018. It can be seen that the present study successfully detected these heatwave events. In 2013, 4–6 heatwaves were detected in the Yangtze River Basin (1–4 in other years) (Figure 10a), the HWMHI values reached 18–30 (Figure 10c), the longest heatwaves lasted 10–20 days, and a heatwave lasting 23 days occurred in Central and North Hunan (Figure 10e). The heatwaves in Northeast China in 2018 mainly occurred in two regions: the junction of Central South Heilongjiang and Northeast Jilin and the junction of Liaoning and Southeast Jilin (Figure 10b,d,f).

The present study showed an obvious upward trend in the northeast and the Qinghai–Tibet Plateau during TI, which aligns with the results of previous studies [81,82,83,84]. We further verified the reliability of the results by comparing them with those of past research [49], as shown in Figure 11. The results revealed good consistency in the eastern, northeastern, and northwestern regions of China, and high-hazard areas, such as Southeast Tibet, East Xinjiang, Chongqing, and other regions, were accurately identified. However, the results obtained for South China, such as the Hainan and Guangxi regions, were quite different, which means that Yin’s results may have overestimated the hazards in these areas. The possible reason is that the definition in the past study only included the annual relative temperature threshold and not the long-term changes. In addition, another possible reason is that Yin’s study involved more tropical and arctic regions, which reduced the differences within China. Third, to make the results comparable between years, it is also necessary to normalize the annual indicators using a unified threshold. In contrast, the results of the heatwave hazard assessment in this study are more consistent with those of existing studies. They are also more accurate and better reflect the hazard levels in China.

Several studies have shown that urban heat islands and heatwaves synergistically increase the overall risk levels of cities [85,86]. In 2017, the China Meteorological Administration released the rankings for hot summer cities in China, which included Chongqing, Fuzhou, Hangzhou, Nanchang, Changsha, Wuhan, Xi’an, Nanjing, Hefei, and Nanning. Therefore, these are known as the “ten furnaces” [87]. A similar conclusion was drawn in the present study about the new “ten furnaces”, as mentioned above. While the previously reported results differed slightly from ours, they were similar overall. Specifically, seven of the originally reported cities, including Nanchang and Wuhan, remained in the “ten furnaces” group identified in the present study, while Nanning, Xi’an, and Fuzhou were replaced by Zhengzhou, Shijiazhuang, and Haikou. This may have been due to the differences in the research scale (point scale and grid scale) and the calculated standards (HI and heatwave hazard). Zhengzhou, Wuhan, Changsha, and Nanjing were found to have experienced significant increases in hazard levels. Previous research has pointed out that the North China Plain is threatened by deadly heatwaves [44], which provides another explanation for the cities in North China becoming “furnace cities.” Notably, Jinhua (Zhejiang) ranked first in terms of AH values, and this result was similar to past research [57]. The possible reason for this is that Jinhua is surrounded by mountains and experiences rapid warming during the day and slow cooling at night [88]. Research shows that defining city classes, which was the method used in this study, can help focus on cities that are easily overlooked in heatwave research.

### 4.2. Strengths and Limitations of the Study

Previous studies have pointed out that extreme regional heatwaves will increase significantly with global warming [89,90,91], while temporal or spatial compounding heatwave events will further increase these heat-related hazards [62,92,93]. The traditional hazard assessment method of heatwaves only considers temperature variables, which may lead to inaccuracies in dry and wet areas. Compared to studies that only consider temperatures, the present study can better reflect people’s feelings. In addition, the hazard assessment of China’s heatwaves at the grid scale, particularly with a spatial resolution of 0.01°, was more refined than those of meteorological stations, and this method can help intuitively describe the heatwave grades in different regions. Third, the heatwave research tends to focus more on cities due to the impacts of urban heat islands [46,85,86]. However, a large number of farmers and migrant workers in vast rural China, who are relatively poor and need to work in hot weather for long periods of time, also need to be considered. This study’s results showed spatial continuity, as they relate to whole areas, and may be used to provide assistance to the inhabitants of these areas from a heatwave perspective.

However, it should be noted that the relationship between heatwaves and human health is extremely complex, so the apparent temperatures used in different studies are not unique. In some cases, the adverse effects on humans may have been overestimated (such as the narrow tube effect between large high-rise buildings) or underestimated (such as strong solar radiation outdoors) [94], but the effects of radiation and wind speed on the human body are relatively insignificant. Therefore, the indicators selected for this study did not consider the effects of the wind speed or underlying surface. The effects of the apparent temperature were relatively insignificant [47]. In addition, the importance of each indicator was not differentiated, which is also a limitation of the study, because different regions and different indicators have different effects on hazards.

### 4.3. Implications for Future Heatwave Research and Public Policy

Previous studies have shown the additional mortality caused by an initial heatwave equals 5.04%, which is higher than the mortality rate afterwards (2.65%) [95]. Heatwaves affect the thermoregulatory system of organisms through heat stress, causing harm to organisms and the natural world [96]. Previous studies have shown that the faster the temperature changes, the higher the heat stress [97,98]. Therefore, it is necessary to pay extra attention to heatwave characteristics such as heatwave start and end dates and different heatwave grades. These characteristics can also provide us with a heatwave’s spatial and temporal information. In fact, the grid datasets for TI and HI values in China are convenient for heatwave characteristics research.

Several areas highlighted in Figure 6 need our attention, including Southeast Tibet, East Xinjiang, Chongqing, North Henan, and Central Zhejiang. The values of the heatwave indicators in Chongqing and Central Zhejiang were much higher than the national averages, and the populations in these areas are growing due to economic construction and tourism, raising the need to pay more attention to these regions. The government should rationally determine the schedules of outdoor workers (such as sanitation workers and construction workers) based on the heatwave characteristics of the different regions, while other social organizations should actively publicize relevant heatwave prevention knowledge to reduce the possibility of residents being exposed to high temperatures and humidity levels for extended durations, thereby reducing the possibility of heatwaves harming their health. In addition, several areas with low spatiotemporal hazard levels were identified in this study. Because their high sensitivity to climate change and high ecological vulnerability, the ecological protection and construction of these areas should not be neglected.

By accurately forecasting heatwave weather, potential deaths due to heatwaves can now be avoided via the joint efforts of the government, individuals, and caregivers of at-risk people [99]. Previous studies have shown the success of heatwave warning systems, which can result in a large amount of people being saved, as well as high profits [99,100]. However, social heatwave handling systems need to be further researched. The demand for electric power will be difficult to meet under the combined pressure of soaring electricity use and drought, which is more prevalent in the southern regions of China during heatwaves, as these areas rely more on hydropower. The present study’s results also highlighted some of these regions. In addition, meteorological disasters, such as dry–hot winds and forest fires caused by heatwaves, will also have impacts on agriculture and forestry, leading to the loss of human society. In severe cases, these will threaten human health and even human lives. If a more comprehensive assessment of heatwave hazards is required in the future, the indicators that directly or indirectly pose health threats to humans should be focused on.

## 5. Conclusions

Based on the interpolated MT and RH data, we calculated the TI and HWI values. Based on the HWI, the HWF, HWMHI, and HWMD datasets for Chinese mainland, covering the years 1990–2019, were obtained with a spatial resolution of 0.01°, and their spatiotemporal distribution was analyzed. Then, the spatiotemporal distribution of the heatwave hazards was evaluated. The results revealed the following points.

(1)The TI increased in 1990–2019, albeit with fluctuations, with the highest ATIRC value found in North China, followed by Northwest China, in July (3.39%). Notably, there was a clear trend of increasing TI values in climate-sensitive regions, such as Northeast China and the Qinghai–Tibet Plateau, in June and September.(2)The areas with medium hazard levels and above were mainly distributed in East and South China, Southeast Tibet, East and South Xinjiang, and Chongqing, accounting for 22.16% of the total. The areas with significantly increasing hazard levels were East China, South Xinjiang, and Western Inner Mongolia. Through a comparative analysis, the areas with “high and rapidly increasing” hazard levels, such as Southeast Tibet, South Xinjiang, Chongqing, South Hebei, West Henan, Central Zhejiang, Central and South Jiangxi, and East Hunan, accounted for 8.71% of the country, while the areas with “low and continually increasing” hazard levels were widely distributed in the eastern, southern, and northern regions of China, including Jiangsu, Inner Mongolia, Hainan, Shandong, and Heilongjiang, which accounted for 41.33% of the total.(3)The city with the highest AH value was Luohe (Henan), while the city with the fastest growth was Suzhou (Jiangsu). The units of cities and counties were found to have increased significantly by 57% and 68%, respectively. Among the 49 first-tier, new first-tier, and second-tier cities, the top 10 were Jinhua, Zhengzhou, Nanchang, Wuhan, Shaoxing, Changsha, Shijiazhuang, Nanjing, Wuxi, and Changzhou. Some of these cities have low administrative or economic development levels, which have reduced the attention paid to the mentioned cities. However, it is necessary to pay attention to the internal infrastructure construction in these cities to reduce the harm of future heatwaves. Upon ranking the provincial capitals, the “ten furnaces” were identified as Zhengzhou, Nanchang, Wuhan, Changsha, Shijiazhuang, Nanjing, Hangzhou, Haikou, Chongqing, and Hefei (sequentially).

## Figures and Tables

**Figure 1 ijerph-20-01532-f001:**
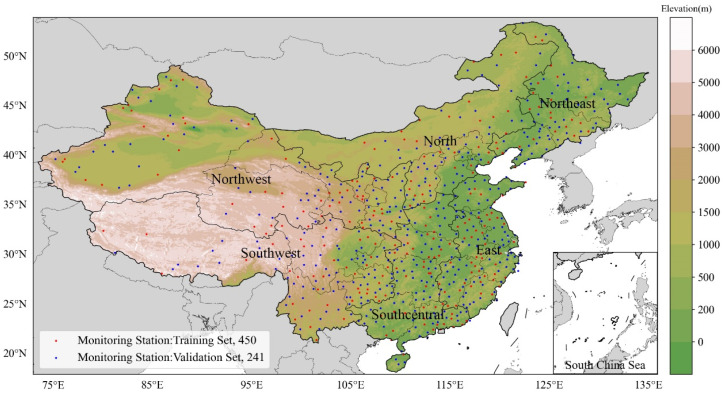
Domains of six geographical divisions—north (N), which consists of Beijing, Tianjin, Hebei, Shanxi, and Inner Mongolia; northwest (NW), which consists of Xinjiang, Qinghai, Gansu, Ningxia, and Shaanxi; northeast (NE), which consists of Heilongjiang, Jilin, and Liaoning; southwest (SW), which consists of Tibet, Yunnan, Sichuan, Chongqing, and Guizhou; south central (SC), which consists of Henan, Hubei, Hunan, Guangdong, and Guangxi; and east (E), which consists of Shandong, Anhui, Jiangsu, Shanghai, Zhejiang, Jiangxi, and Fujian—as well as meteorological stations. Four sets of random validation stations were selected to make their distribution more uniform. Containing 10% of all stations (of a total of 69), each set randomly selects one day of the month to apply, for a total of four days per month using only interpolation points for interpolation.

**Figure 2 ijerph-20-01532-f002:**
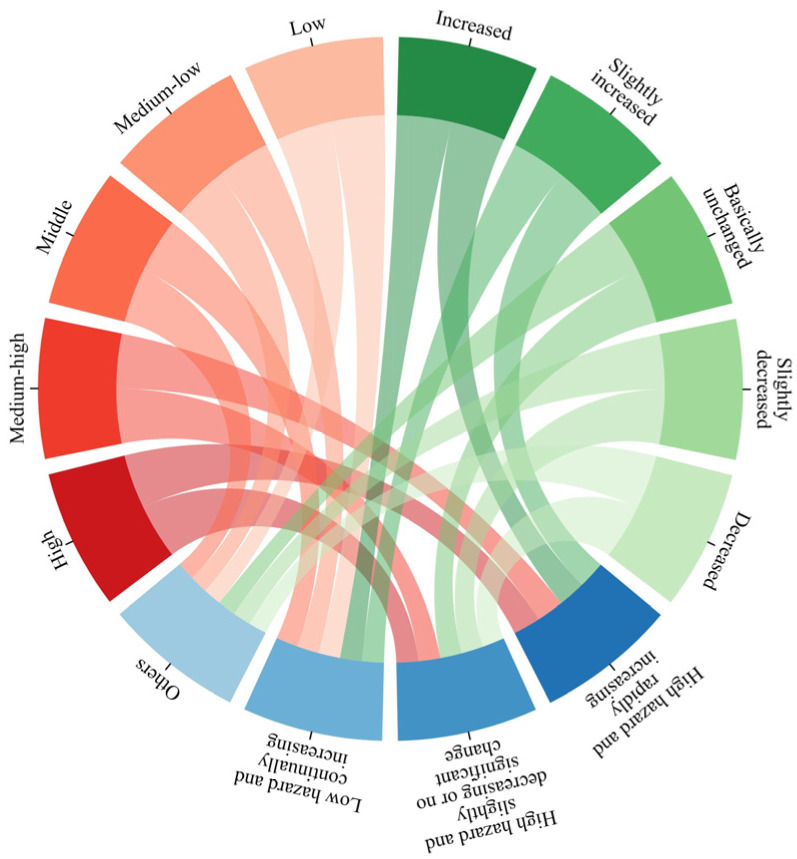
The standard for spatiotemporal classifications of heatwave hazards. The red and green sections represent the spatial distribution and temporal change classifications of heatwave hazards, respectively, while the blue section represents the spatiotemporal classification of heatwave hazards.

**Figure 3 ijerph-20-01532-f003:**
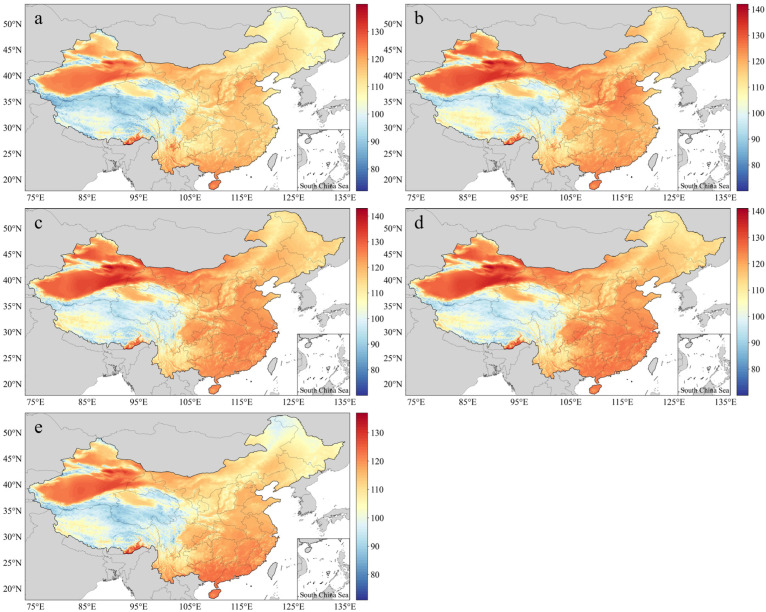
Spatial distribution of annual average TI (ATI) values for the years 1990–2019: (**a**) May, (**b**) June, (**c**) July, (**d**) August, (**e**) September.

**Figure 4 ijerph-20-01532-f004:**
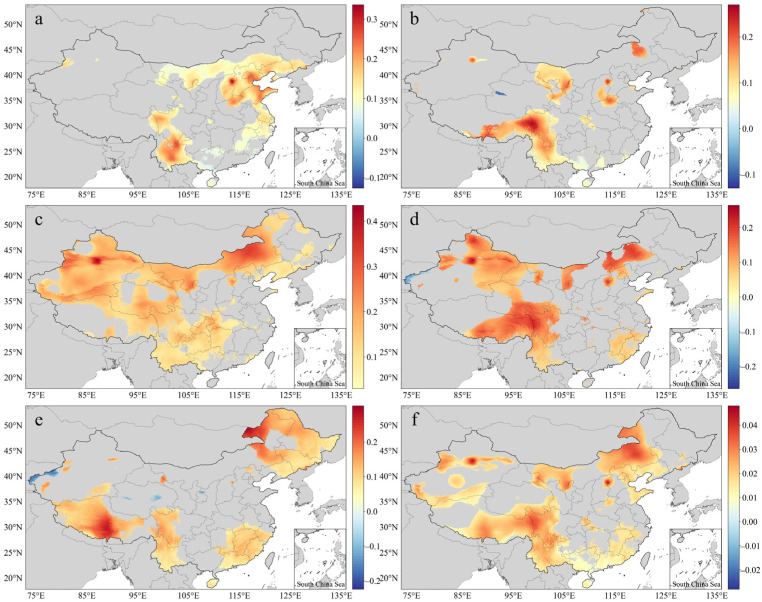
Slopes of the mean TI: (**a**) May; (**b**) June; (**c**) July; (**d**) August; (**e**) September; (**f**) monthly.

**Figure 5 ijerph-20-01532-f005:**
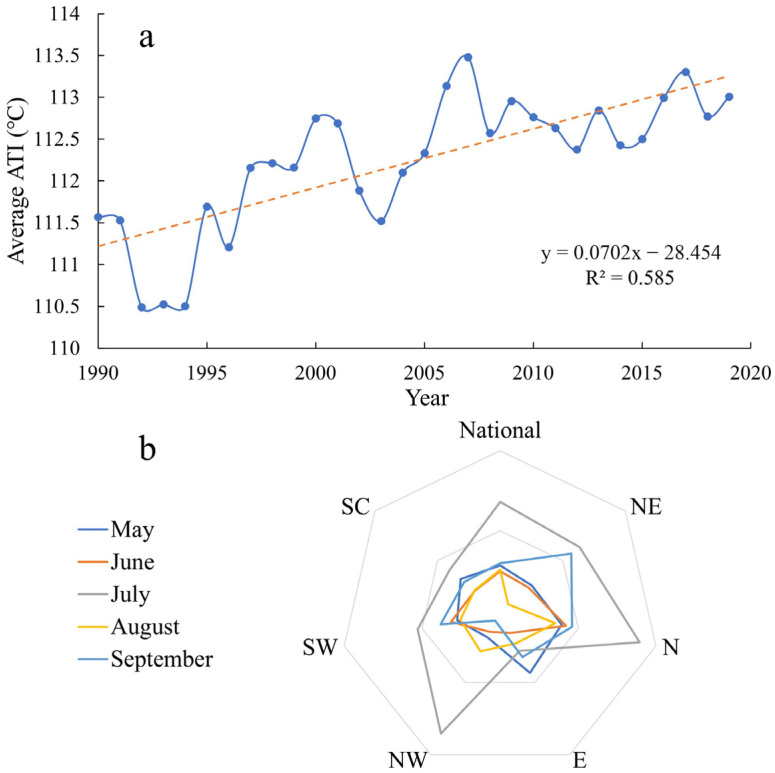
Annual changes in the national ATI in 1990−2019 (**a**) and the RC values of the ATI between 1990–1994 and 2015–2019 (**b**).

**Figure 6 ijerph-20-01532-f006:**
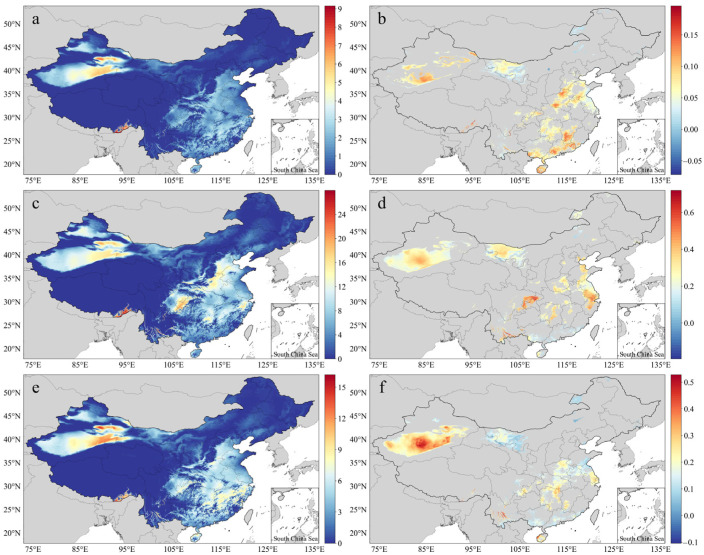
Spatial distribution of the averages and slopes of the HWF (**a**,**b**), HWMHI (**c**,**d**), and HWMD (**e**,**f**) for the years 1990–2019.

**Figure 7 ijerph-20-01532-f007:**
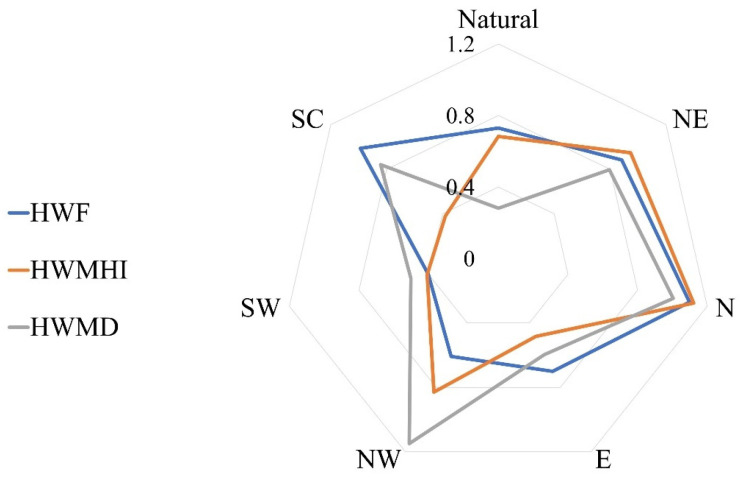
RC values of HWF, HWMHI, and HWMD.

**Figure 8 ijerph-20-01532-f008:**
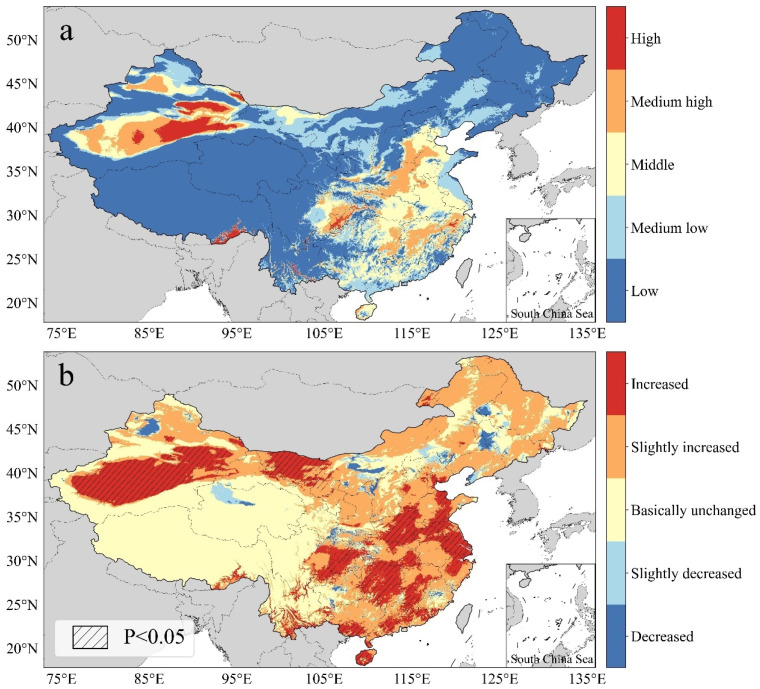
Heatwave hazard assessment (**a**) and slope of the AH values (**b**).

**Figure 9 ijerph-20-01532-f009:**
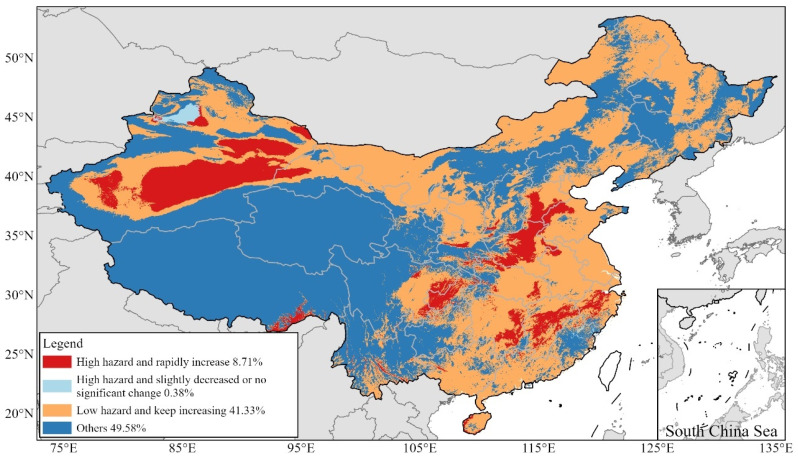
Heatwave classifications in China.

**Figure 10 ijerph-20-01532-f010:**
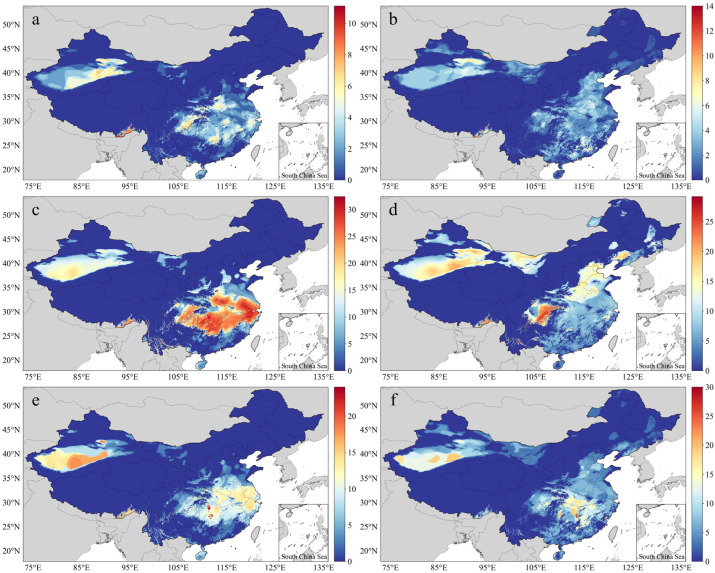
HWF, HWMHI, and HWMD values in 2013 (**a**,**c**,**e**) and 2018 (**b**,**d**,**f**).

**Figure 11 ijerph-20-01532-f011:**
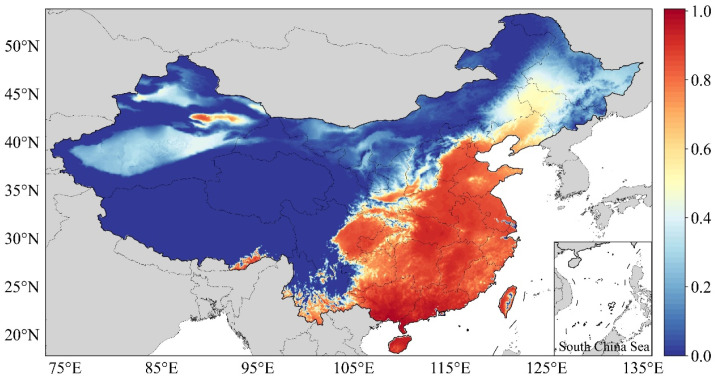
Heatwave hazard assessment. Data from Yin’s results [49].

**Table 1 ijerph-20-01532-t001:** Interpolation verification results.

Date	Variable	Slope	R^2^	P	MAE	NMAE	RMSE	NRMSE
22 May 1994	RH	0.9883	0.9585	1.2 × 10^−47^	3.6425	0	4.8385	0
5 May 2002	RH	0.9906	0.9393	8.9 × 10^−43^	4.4610	0.6736	5.6748	0.4558
22 May 2010	RH	0.9853	0.9390	1 × 10^−42^	4.4661	0.6777	5.8611	0.5574
22 May 2016	RH	0.9855	0.9427	5.25 × 10^−43^	3.9316	0.2379	5.3401	0.2734
May 1997	RH	0.9940	0.9078	4.58 × 10^−142^	4.8122	0.9625	6.3506	0.8241
May 1999	RH	1.0043	0.9064	3.53 × 10^−141^	4.6614	0.8384	6.4561	0.8816
May 2016	RH	0.9981	0.9156	8.96 × 10^−148^	4.6932	0.8646	6.5814	0.9499
1997	RH	0.9933	0.8652	0	4.8577	1	6.6733	1
1990	RH	0.9945	0.8645	0	4.6707	0.8461	6.4122	0.8577
17 August 1990	MT	1.0033	0.9717	4.19 × 10^−53^	0.8313	0	1.0669	0
10 May 1995	MT	1.0051	0.9654	2.98 × 10^−50^	0.9118	0.2746	1.3134	0.4205
31 July 2012	MT	1.0081	0.9664	2.17 × 10^−51^	0.9017	0.2399	1.1743	0.1831
27 September 2014	MT	0.9985	0.9659	3.55 × 10^−51^	1.0061	0.5960	1.3341	0.4559
September 1992	MT	0.9996	0.9532	6.97 × 10^−182^	1.0220	0.6503	1.4879	0.7182
September 2003	MT	1.0012	0.9544	4.55 × 10^−184^	1.0433	0.7229	1.5320	0.7934
September 2019	MT	1.0023	0.9486	1.24 × 10^−177^	0.9876	0.5330	1.4638	0.6772
1990	MT	0.9988	0.9409	0	1.1018	0.9227	1.6530	1
2003	MT	0.9994	0.9433	0	1.1245	1	1.6410	0.9796

**Table 2 ijerph-20-01532-t002:** AH rankings of first-tier, new first-tier, and second-tier cities.

Index	City	Index	City	Index	City	Index	City
1	Jinhua, Zhejiang	14	Chongqing	27	Zhongshan, Guangdong	40	City of Yantai
2	* Zhengzhou, Henan	15	Hefei, Anhui	28	* Nanning, Guangxi	41	Guiyang, Guizhou
3	Nanchang, Jiangxi	16	* Xuzhou, Jiangsu	29	* Nantong, Jiangsu	42	* Xiamen, Fujian
4	* Wuhan, Hubei	17	* Suzhou, Jiangsu	30	Taizhou, Zhejiang	43	Taiyuan, Shanxi
5	Shaoxing, Zhejiang	18	* Foshan, Guangdong	31	Beijing	44	Harbin, Heilongjiang
6	* Changsha, Hunan	19	Jinan, Shandong	32	* Huizhou, Guangdong	45	Dalian, Liaoning
7	Shijiazhuang, Hebei	20	Tianjin	33	Chengdu, Sichuan	46	Lanzhou, Gansu
8	* Nanjing, Jiangsu	21	* Yangzhou, Jiangsu	34	Fuzhou, Fujian	47	Changchun, Jilin
9	* Wuxi, Jiangsu	22	Dongguan, Guangdong	35	Wenzhou, Zhejiang	48	* Kunming, Yunnan
10	* Changzhou, Jiangsu	23	* Shanghai	36	Shenzhen, Guangdong	49	* Quanzhou, Fujian
11	* Jiaxing, Zhejiang	24	* Guangzhou, Guangdong	37	Shenyang, Liaoning		
12	Hangzhou, Zhejiang	25	Ningbo, Zhejiang	38	Zhuhai, Guangdong		
13	Haikou, Hainan	26	Xi’an, Shaanxi	39	Qingdao, Shandong		

Note: * means that the city passed the significance test.

## Data Availability

Maximum temperature (MT, °C) and average relative humidity (RH, %) data from 699 meteorological stations in China for the years 1990–2019 were used to calculate the TI and HI values; these formed the surface climate daily value dataset (V3.0) from the China Meteorological Data Network (http://data.cma.cn/, accessed on 13 January 2023). Due to the large amount of data, the gridded dataset (0.01°) interpolated and processed by the authors includes daily TI and HI values, while the temporal and spatial characteristics of annual heatwave indicators and heatwave hazards will be shared in an appropriate manner. The figures were created mainly using Python, ArcGIS Pro, and Excel; the script files related to this manuscript have been uploaded to GitHub (https://github.com/wuming365/china-contourf, accessed on 13 January 2023).

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
