# Peer review of "Spatiotemporal Distribution of Heatwave Hazards in the Chinese Mainland for the Period 1990–2019"

_ijerph, 2023, doi:10.3390/ijerph20021532_

Round 1

Reviewer 1 Report

The method section is too long.

The discussion section is too long.

Abstract, page 1, line 16 and 17, merge two short sentences “TI rose but with fluctuations. The largest increase occurred in North China in July.

Introduction, page 3, line 126, merge this sentence (the data were quality controlled) with the previous sentence.

Page 5, the title of table 1 is missing.

Section 2.4.1, page 5, “HI” introduced in introduction. Remove “heat index”, also line 203 about “HWI”, and page 6, line 254 for AH. Check all of abbreviations.

Page 7, line 260, “figure 2 presents the classification standards”. This sentence is not complete. Rewrite this sentence.

Page 8, line 313, This sentence is too short. You could merge it “A significance test was performed. ”.

Page 12, Correct the table number. Table 2 is correct.

Page 13, line 429, What is “mu”?

Page 13, lines 431, 434, 435, 436, 439 and 441, what is “Error! Reference source not found”? Find and correct in the whole of the manuscript.

Page 14, the caption of figure 11, “(Yin et al.)” The year is missing.

Page 15, line 508, this word “somatosensory” is not appropriate. Do you mean “feeling temperature”?

Reviewer 2 Report

If the method of this paper is innovative enough, it is suggested to revise and publish.

The main doubts are whether the method is innovative and reasonable. Since it is mentioned in the paper that " a heatwave hazard is a measure of the severity of heatwave events, usually determined by the intensity, duration, frequency, and extent of heatwaves (line 72-73)", " it is rather unreasonable to judge a heatwave in a complex climate region based on a single absolute threshold or a simple relative temperature threshold (line-60)". This study can better reflect people's feelings (line-495). So in China, a place with a complex climate zone, does the use of "maximum temperature" and "humidity" data, and the equal weight given to "frequency, intensity and duration of heat waves" simply consider the composition of heat wave hazards? How do I prove that the 2.8 threshold does not repeat the "unreasonable" problem described earlier? In what way is the index "better reflective of people's feelings"? Since many scholars have studied the temporal and spatial distribution of heat wave occurrence in China, please emphasize the innovation or advantage of the method used in this paper.

1. Introduction

It is suggested to add the reasons why HI and TI are selected as WHI index in the introduction.

2. Materials and Methods

Line-208: How applicable is this threshold in China? What is the basis for the division?

Line-240: What are the considerations for normalization? What is the weight of HWF\HWMD\HWMHI?

Line-241: Formula insertion error

It is suggested to streamline the Materials and Methods section to make it more readable.

3. Results

Line-305: "Affected by high relative humidity and low wind speed, the southeastern provinces of China, Sichuan, and Chongqing reached maximum Tl levels and coverage in August." This sentence cannot be drawn from this study.

4. Discussion section

Line-431: Reference insertion error.

Line-461: "and it is unreasonable to simply normalize them to 0-1, which is another issue in his study. In contrast, the results of the heatwave hazard assessment in this study are more consistent with those of existing studies; they are also more accurate and better reflect the hazard levels in China. " Please briefly demonstrate the reasons why the method used in this paper is more reasonable and whether it is supported by other literature.

Round 2

Reviewer 3 Report

My comments have been addressed and the manuscript has been improved, therefore, I recommend it to be published as it is

Author Response

Thank you for your suggestions. According to your advice, this manuscript was edited for proper English language, grammar, punctuation, spelling, and overall style by one or more of the highly qualified native English-speaking editors at Scribendi (https://www.scribendi.com). And the revised manuscript can be seen in the attachment.